# Predicting Event Memorability from Contextual Visual Semantics

**Qianli Xu**
Institute for Infocomm Research
A*STAR, Singapore
qxu@i2r.a-star.edu.sg

**Fen Fang**
Institute for Infocomm Research
A*STAR, Singapore
fang_fen@i2r.a-star.edu.sg

**Ana Garcia del Molino**
ByteDance AI Lab
Singapore
a.g.delmolino@gmail.com

**Vigneshwaran Subbaraju**
Institute of High Performance Computing
A*STAR, Singapore
vigneshwaran_subbaraju@ihpc.a-star.edu.sg

**Joo Hwee Lim**
Institute for Infocomm Research, A*STAR
Nanyang Technological University, Singapore
joohwee@i2r.a-star.edu.sg

## Abstract

Episodic event memory is a key component of human cognition. Predicting event memorability, *i.e.*, to what extent an event is recalled, is a tough challenge in memory research and has profound implications for artificial intelligence. In this study, we investigate factors that affect event memorability according to a cued recall process. Specifically, we explore whether event memorability is contingent on the event context, as well as the intrinsic visual attributes of image cues. We design a novel experiment protocol and conduct a large-scale experiment with 47 elder subjects over 3 months. Subjects' memory of life events is tested in a cued recall process. Using advanced visual analytics methods, we build a *first-of-its-kind* event memorability dataset (called *R3*) with rich information about event context and visual semantic features. Furthermore, we propose a contextual event memory network (CEMNet) that tackles multi-modal input to predict item-wise event memorability, which outperforms competitive benchmarks. The findings inform deeper understanding of episodic event memory, and open up a new avenue for prediction of human episodic memory. Source code is available at `https://github.com/ffzzy840304/Predicting-Event-Memorability`.

## 1   Introduction

Episodic event memory is a key component of human cognition and intelligence. Maintaining high-level event memory is good for mental health and executive functioning [34]. Episodic memory can be enhanced by cognitive training, such as regular photo review that triggers the reactivation of certain memory traces [25, 42]. Recent advancements in visual lifelogging, including affordable hardware and powerful visual analysis software, make it possible to implement lifelog-based memory intervention [19, 31, 43]. However, without clear understanding of what factors influence event memory (*e.g.*, what events have higher value for training? how to select "good" photo cues to boost training effect?), it is difficult to design effective cognitive intervention programs. This study attempts to understand event memory, *i.e.*, to what extent an event can be recalled. We probe the recall process with visual cues, and explore factors that influence event memorability. We aim to extend the understanding of event memory, beyond simple image recognition [22] and memory of event categories [8], to the prediction of item-wise event memorability using contextual visual semantics.

A few major gaps exist in this field. First, there is scarcity of data for evaluating event memorability. A sensible analysis requires rich event knowledge, including spatial-temporal information, personal experience and profiles, as well as quantifiable memory performance on the respective events.

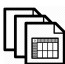

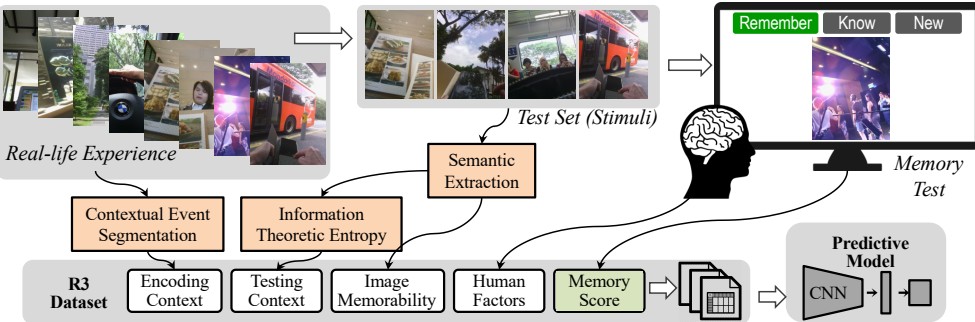

Figure 1: Overview of event memorability prediction.

Considering that life events are highly personalized and seldom repeatable, it is difficult, if not impossible, to collect such data in laboratories by replicating a controlled process. This makes the generation of large dataset for event memorability study a major challenge. Second, event memory involves a complicated cognitive process in the human brain, so that the mapping relationship between event information and the memory outcome is obscure and fuzzy. It is affected by a number of factors, such as personal experience, cognitive status, and new, potentially conflicting input. Therefore, the memory outcome often has immense variability, making it extremely difficult to be predicted reliably.

This study fills these gaps in three aspects. First, we develop a novel experiment protocol to collect personal life event information along with the human subjects' memory outcome on selected events (Figure 1). We build a *first-of-its-kind* event memorability dataset consisting of $10,654$ valid samples with rich information about event context and visual semantic features. Based on data collected in a large-scale user study with 47 elder subjects, each in three months, we apply a suite of visual analysis methods, such as contextual event segmentation (CES) and information-theoretic entropy analysis, which collectively extract visual semantic features and contextual information. Second, we provide evidence that event memorability is contingent on extrinsic, contextual information, as well as intrinsic visual attributes. Third, inspired by image memorability studies [45], we develop a simple yet effective contextual event memory network to predict event memorability by combining multi-modal visual semantics, including the image cues and multiple aspects of contextual information. The model predicts item-wise event memorability with reasonable accuracy and serves as a baseline for more sophisticated methods. The dataset and baseline predicative model establish the foundation for developing computational models to decode human episodic memory.

## 2 Related Work

Episodic event memory has been intensively studied in cognitive psychology, exploring the human behaviors, cognitive processes and neural mechanisms of memory [11, 40]. Regarding factors that influence how much an event is remembered, most studies focus on the macro-level, collective characteristics of events that influence subsequent memory. For example, distinctive events are remembered better than routine ones [21]; episodic details are gradually lost with time; memory could be distorted by interfering cues that re-activate alternative neural processes [25]. Nevertheless, little is reported on the micro-level properties of events that contribute to item-wise event memorability. We believe this issue is important because it opens up new possibilities of computational intelligence in multiple domains such as image retrieval [30, 50], cognitive intervention [8, 51] and diagnosis [2]. Recent works in neuropsychology resort to brain imaging to decode the memory of individual events [3, 5, 39]. However, the brain imaging data is prohibitively difficult to collect, requiring special equipment and controlled environments, which restricts its application in the daily life.

Meanwhile, there is a sizeable body of research that investigates image memorability. It is generally accepted that memorability is an intrinsic property of images based on the recurring evidence that human subjects show considerable consistency in image recognition tests [4, 23, 22]. Accordingly, computational models have been developed to predict image memorability from the visual semantic information, which is partially made possible by high-quality image memorability datasets, such as, SUN-Mem [22], LaMem [28], FIGRIM [8], Mem-Cat [17], LNSIM [33], *etc*. Leveraging on deep convolutional neural network (CNN) and well-crafted content analysis mechanisms, recent

methods have achieved superior performance in image memorability prediction [15, 45, 26, 33, 29]. Some works explore the role of extrinsic factors in image memorability, such as, eye gaze [1], testing context [8, 37], image organization [16], emotion [6] and various external sources [26]. It is shown that these extrinsic factors are conducive to the prediction of image memorability, although their predicative power is arguably lower than the intrinsic factors [7, 32, 41].

If we consider image viewing and recalling as a 'mini-event', it is tempting to apply knowledge of image memorability to that of events, especially in view that event memory experiments often resort to image cues to trigger the memory. However, the memory of a static image is not equivalent to that of an event. The latter involves much more complicated cognitive processes and is affected by many internal (event properties) and external factors (*e.g.*, event context and personal characteristics). Therefore, it is much more difficult to predict event memorability at the micro-level in the real-world context. Video memorability is another stream of closely related works [36, 10]. Continuous video clips show events with temporal order, which may resemble episodic events. Nevertheless, there are fundamental differences between third-person-view video memory and the first-person-view (FPV) real-life event memory, namely, real-life events are of personal relevance, strongly contextualized in a myriad of life events. As such, video memorability still lacks the rich context of personal experience. Foremost, there is no dataset of event memorability that accounts for the multivariate factors in its prediction. While cued-recall has long been adopted in autobiographical memory studies (*e.g.*, using lifelog photos), existing works mostly focus on the behavioural and neuropsychological processes [25, 3], rather than computational prediction of event memorability. To fill these gaps, this study will develop a novel dataset with rich information of event memory and propose a new baseline model for its prediction.

## 3 Dataset

### 3.1 R3 Experiment

We design an experiment called R3 (standing for *record, retrieve, replay*) to investigate event memorability in the real-world context. Forty-seven elder subjects (37 female, age: mean = 62.7 years, std = 6.3 years) participated the study over three months. During the first 4 to 5 weeks, they used a wearable camera (Narrative Clip, temporal resolution set as 2 frames per minute) for about 7 hours per day to collect lifelog data. Twice a week, they visited the experiment site to upload the collected lifelog data and underwent scheduled treatment. In a treatment session, they were shown lifelog photos selected from the past few days (*i.e.*, since last visit), and were guided to make an effortful recall of the related events, in the same vein as the episodic specificity induction method [34]. Half of the site visits involved treatment and the other half did not, leading to a balanced experiment control for studying the treatment effect. Interested readers may refer to supplementary material and [52] for detailed information of the experiment protocol.

Subjects were tested for their autobiographical memory twice in the first month, each with two weeks' data collection and four visits to the experiment site (including two treatment and two non-treatment sessions). In each memory test, a set of 144 distinctive photo cues were presented one at a time for the subject to recall and report the respective memory strength. Among the 144 cues, 1/3 were sampled from the treatment period, 1/3 from the non-treatment period, and 1/3 were lure photos (*i.e.*, not from the subjects' own life). The lure photos served a similar purpose as the "filler" as in a standard image memorability test [23]. Subject were aware of the existence of lure images, through pre-test briefing and warm-up trials.

For event memory testing, a photo cue was shown for 6 seconds, after which a subject responded by rating the respective event memory according to the "Remember-Know" paradigm [40], which is widely adopted in psychology studies. In particular, subjects rated their memory of the cued-event in two questions: (1) type of memory, which could be one of *remember* (recollection with certain episodic detail), *know* (familiar but no episodic detail), or *new* (not from 'my' life); (2) memory level, which refers to the degree of the respective memory type, including recollected event detail 1–4 (for *remember*), familiarity level 1–4 (for *know*), and confidence 1-2 (for *new*) [52]. This resulted in a graded memory strength ranging from 9 (recollection with a lot of episodic details), to 5 (very familiar but without episodic details) [46], until 0 (no memory at all, *i.e.*, the cued-event was considered absolutely not from 'my' own life). Readers may refer to [39] for a similar mnemonic scheme. After the second test, subjects did not record any more lifelog data. However, a third test was arranged two months later using photo cues drawn from the entire lifelog repository, but not used in the treatment,

nor previous two tests. This was intended to evaluate the long-term memory outcome. A total of 288 photos (from subjects' own life) were rated with a memory strength from three tests, among which 2/3 (from test sessions 1 and 2) had a short encoding-testing interval (2.5 days on average), and 1/3 (from test 3) had a longer encoding-testing interval (10 weeks on average).

The main outcome of the R3 experiment are (1) a lifelog repository from 47 subjects, each with an average of $25,000$ photos captured in about 30 days, and (2) about $16,000$ photo cues (excluding lure photos) with memory scores that indicate the strength of the event memory. As a mechanism of quality control, data samples were excluded if they met the exclusion criteria (refer to A.1.4). In total, $10,654$ photos cues from 40 subjects were included in the dataset. Each photo has a memory score of $0 \sim 9$, serving as the ground truth. The photo cues were further processed to extract (1) visual semantic features and image memorability, (2) contextual information related to event encoding, and (3) contextual information related to memory test (Figure 1), which will be elaborated next.

## 3.2 Intrinsic Factors

This study uses selected FPV photos as visual cues to trigger event memory. Considering that the intrinsic attributes of images cues may affect the memorability of the denoted events, we account for the intrinsic memorability of visual cues by computing the memorability score directly from visual features. As aforementioned, there is an abundance of methods to do so. We adopt a few representative, pre-trained models, including, MemNet [28], AMNet [15] and DeepNSM [33] (model is re-implemented from the paper). These methods typically account for high-level visual semantics (*e.g.*, object, human face, scene, *etc.*), as well as low-level features, *e.g.*, GIST, SIFT, HOG, and pixel histograms). Some high-level semantic features, including, human face and body, are particularly important in image memorability prediction [15]. However, automated detection algorithms (as embedded in the image memorability models) may not achieve the desired performance. Therefore, we manually annotate the image cues on the presence of human face and body. An image is considered to have human face(s) if one or a few faces are clearly visible. Similarly, if a significant proportion of human torso is visible, an image is considered to contain a human body.

## 3.3 Extrinsic Features

### 3.3.1 Event Encoding Context

Since the events are from subjects' own life, the individual experience may affect how an event is memorized. Memory of an event starts from the encoding (when it occurs). Encoding context refers to how an event is associated with one's life experience. Theories in cognitive research suggest that event distinctiveness [21] and event boundary conditions [18] are important factors of event memory.

1. *Event distinctiveness* refers to how unique an event is relative to other life events. In general, rare events (*e.g.*, attending my daughter's wedding ceremony) have strong and lasting memory traces, while routine events (*e.g.*, going to the neighbourhood grocery store) tend to lose the episodic details over time [21]. To quantify event distinctiveness, we extract the visual semantics of a photo cue, and compare it with those extracted from the lifelog photos belonging to the subject. In particular, we use InceptionV3 [48] pre-trained on ImageNet to extract visual features of an image. The similarity between two images are computed as the cosine similarity of the two visual vectors. We compute the similarity score of a photo cue against an individual subject's lifelog collection. Applying a threshold, we can get the proportion of lifelog photos that are similar to the photo cues. This proportion value thus indicates how 'rare' (*i.e.*, distinctive) the event is. In practice, we apply three thresholds 0.4, 0.6 and 0.7 to get event distinctiveness at varying granularity.

2. *Event boundary condition* refers to the temporal distance of the snapshot of an event (*i.e.*, the photo cue) to the start/end of that event. For example, if a person goes for an excursion, the photo cue may show the start of the journey, the end of it, or the middle part of the event. According to the event segmentation theory (EST) [18], photo cues closer to the start of an event are remembered better than if they are from the middle or end of an event. To evaluate event boundary condition, we need to characterize events from visual semantics. Let $I = i$ denote the entire lifelog photos belonging to a subject. Each photo $i$ potentially belongs to a unique event $e \in E$. To establish the set of events $E$, we adopt the contextual event segmentation (CES) method [14], which segments continuous lifelog photos into events based on an estimation of event boundaries. In particular, each lifelog photo is

evaluated for its visual context by comparing it with the past and future photo-streams. An autoencoder based on Long Short-Term Memory (LSTM) is trained, leveraging on the CNN image descriptor. This results in a boundary function $b_i = B(i)$, where $B(i) \mapsto [0, 1]$ computes a boundary score that indicates the degree to which $i$ is located at the boundary of an event. Applying a boundary threshold $b_T$, if $b_i \geq b_T, b_j \geq b_T$ and $b_k < b_T$ (for $i < k < j$), the entire set of images between $i$ and $j$ constitutes event $e$. Note that one can control the granularity of events by applying different thresholds. In this study, we test different thresholds: 0.75, 0.9 and 0.95. This results in average numbers of daily events of 16.5, 9.6 and 6.3, respectively. In practice, an average of 10 events per day is considered normal [24, 39], indicating a threshold of 0.9 is appropriate. Therefore, we adopt this threshold in subsequent analysis. Nevertheless, we include event boundary conditions from all three thresholds in the dataset for potential exploratory study. Based on the above event characterization using CES, the photo cues used in the memory test are backtracked to the original events and the temporal distance to the start of the event is recorded. Furthermore, the temporal span of the respective events are recorded to account for event duration.

3. *Activity*: It refers to what activity the subject was performing when the lifelog photo was taken. It is assumed that different activities may involve varying levels of memory retention [50]. Considering the lack of robust methods to recognize activities from FPV images, we manually annotated the photo stimuli according to 33 activity types (extended from [49]). The entire list of activities are available in A.3.

4. *Place*: It is known that different places/scenes lead to varying levels of image memorability [33, 50]. To precisely capture the place where an event occurs, we provide place information of the testing cues. Existing CNN models (e.g., [20]) provide limited accuracy on place recognition, and importantly, they only give general place categories (e.g., Place365 [53]). To make the scene information relevant and accurate, we manually annotate the images according to 45 place categories. The full list of places is available in A.3.

Both place and activity annotations were conducted by two human annotators and the inter-rater reliability was high with Cohen's kappa $\kappa_{activity} = 0.73$ and $\kappa_{place} = 0.77$, respectively.

### 3.3.2 Event Testing Context

Event testing context refers to how a photo cue relates to other cues (including a subject's own photos and lure photos) during the test session. According to [8], contextually distinctive images are more memorable, *e.g.*, if a photo cue is similar to many other testing photos, it would be less distinguishable and may lead to reduced memory strength. We adopted an information-theoretic model to estimate the testing context [8]. A testing context $C$ is considered to be the entire set of 144 photo cues in a test session. Each image $i \in I$ is represented as a feature vector $f_i = F(i)$, where $F$ is a function for feature mapping. In this research, we apply the InceptionV3 [48] as presented in Section 3.2 to implement $F$. The likelihood that an image appears in the testing context is computed as:

$$P(f_i) = \frac{1}{||N_C||} \sum_{j \in N_C} K(f_i - f_j) \tag{1}$$

where $K$ is a kernel function (implemented as an Epanechnikov kernel), and $||N_C||$ is the context size, measured as the number of images in the current test session. According to the present experiment protocol, $||N_C|| = 144$. Next, the distinctiveness of a photo cue is computed as

$$D_C(I) = -\log P_C(f_i) \tag{2}$$

Another important testing context is the *Interval* between event encoding and testing [35], which is simply calculated as the number of days spanning from the lifelog photo recording to the memory test. We anticipate weaker memory strength for longer intervals. Finally, in lined with the effect of memory re-instatement [25, 29], this study used *treatment* to indicate if an event has gone through photo review as a memory intervention mechanism. Treatment is expected to enhance the respective event memory. Other theories related to extrinsic factors, such as, emotion [13, 38], attention [12], are not included because of the difficulties in evaluating them computationally under the current experiment protocol.

| Intrinsic Factors | $t$-statistics | Encoding Context | $t$-statistics | Testing Context | $t$-statistics |
|---|---|---|---|---|---|
| Image memorability | **11.14** | Encode distinctiveness | **7.45** | Test distinctiveness | **3.93** |
| Presence of faces | **10.55** | Boundary condition | -0.73 | Treatment | **10.00** |
| Presence of human | **2.08** | Activities | **7.34** | Interval | **-14.22** |
| | | Places | 1.38 | | |

Table 1: Factors that affects event memorability. $t$-value in bold font means the factor significantly correlated with memory ($p < 0.05$).

## 3.4 R3 Dataset Summary

Through the above operations, we generate the *R3 dataset* consisting of $10,654$ photo cues (extracted from $1.15$ million FPV lifelog photos) with memory scores and rich semantic and contextual features. To the best of our knowledge, this is the first, large-scale dataset that includes rich visual and contextual information associated with event memory from the real-world context.

- Intrinsic visual features: image memorability rating from pre-trained models, including, MemNet [28], AMNet [15], and DeepNSM [33], presence of human faces, presence of human body, and the original photo cues (RGB images).

- Extrinsic context features: (i) encoding context, including event distinctiveness at encoding, event boundary, place, and activity information, (ii) testing context, including distinctiveness at testing, encode-test internal, and treatment conditions, and (iii) subject demographic information. Notably, these features cannot be obtained from individual photo cues.

# 4 Predicting Event Memorability

## 4.1 Understanding Event Memory through Linear Regression Analysis

We hypothesize that event memorability is dependant on both intrinsic and extrinsic features. We first check if image memorability predicts event memorability. We adopt a few representative models that predict image memorability from deep CNN features. For each method, we compute Pearson's product-moment correlation between the ground truth event memory score and the predicted image memorability. MemNet [28] gives a correlation coefficient of $r = 0.02, p = 0.04$, indicating weak correlation between the true event memory and the image memorability. The prediction of DeepNSM [33] is largely uncorrelated with the ground truth with $r = 0.01, p = 0.54$. This is not unexpected because DeepNSM is designed to predict memorability of natural scenes, which are notably different from the lifelog photos in daily life. AMNet [15] gives the best prediction with $r = 0.19, p < 0.001$, showing significant correlation between the two variables. Hence, the intrinsic image attributes do contribute to the prediction of event memory. However, the predictive power is limited, indicating that other extrinsic factors may affect event memorability. Note that in the above evaluation, we use pre-trained models to predict the memorability directly. In Section 5, we implement two competitive CNN models, namely, AMNet [15] and ICNet [45], which are re-trained on the current dataset to boost the predictive power.

Next, we conduct a linear mixed-effect analysis to better understand the contribution of individual features (Table 1). These features are used as fixed effects with "Subject" modelled as a random effect to account for individual intercepts. Among the three intrinsic factors, we use image memorability predicted by AMNet since it gives the best prediction. Table 1 summarizes the $t$-statistics where a larger $t$ indicates a higher impact of the factor on event memorability. From the result, we observe that both intrinsic visual information and extrinsic contextual features contribute to event memorability. The most important features include image memorability, presence of faces, event distinctiveness at encoding, activity, treatment, and encode-test interval.

## 4.2 Predicting Item-wise Event Memory using DCNN

We propose a contextual event memory network (CEMNet) to address the more challenging task of predicting item-wise event memory. The model is built by fusing multi-modal inputs consisting of contextual visual semantics and image data along two pipelines. Pipeline 1 handles the quantified extrinsic and intrinsic (human face and body) features; and pipeline 2 deals with the RGB image input (intrinsic). The structure of the model is shown in Figure 2. For implementation, pipeline 1 adopts a simple multi-layer perceptron (MLP) with three fully connected (FC) layers using "relu", "sigmoid" and "softmax" activation functions, respectively. Pipeline 2 adopts a CNN model to extract image

features, which can be implemented in different fashions, such as, VGG [44], InceptionV3 [48], ResNet [20], *etc*. One may also include an attention mechanism to capture the important regions of the image for refined prediction. For example, we implement AMNet [15] in the CNN pipeline, which adopts ResNet50 and iteratively generates attention maps of potentially important image regions. Each pipeline outputs a feature vector from the respective data source, which are concatenated and passed to an FC layer with "relu" activation function to predict the joint memorability score. Notably, the output of the model, *i.e.*, the memorability score, is an ordinal number instead of categorical data. Therefore, we use ordinal categorical cross-entropy as the loss function [9].

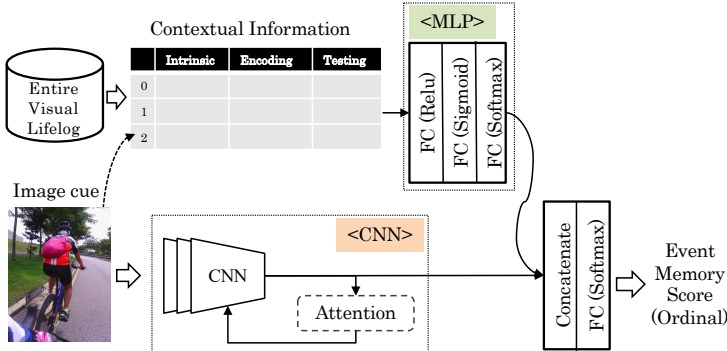

Figure 2: CEMNet Model for item-wise prediction of event memorability.

## 5 Experiment

We compare the performance of different configurations of the event memorability prediction models. First, we randomly split the dataset with a 4:1 train-test ratio based on subject index. Splitting the data with respect to subjects instead of individual image cues is important because it prevents the model to "see" similar data (belonging to the same subject) during the test stage. In the latter case, the experimental results may look good, whereas it does not generalize to unseen users, thus is of little practical value. We perform 5 random splits and evaluate the average performance (*i.e.*, 5-fold cross-validation). We adopt precision, recall, and F1-measure of categorical memory types as the main performance metrics. We first compute the above metrics for individual memory categories on a split. Then, we take the average of 10 categories, before averaging on all 5 splits. Moreover, we report the mean absolute error that is computed as the average distance between predicted and ground truth memory score. The following configurations are evaluated.

- *AMNet* [15]: It is a variant of CEMNet that implements AMNet in pipeline 2, while deactivating MLP in pipeline 1. It is equivalent to re-training AMNet on R3 dataset, serving as a competitive benchmark.

- *ICNet* [45]: ICNet is implemented in pipeline 2, and MLP is deactivated in pipeline 1. It is equivalent to re-training the ICNet model on R3 dataset.

- *MLP*: Only pipeline 1 in CEMNet is activated during training so that image visual features (pipeline 2) are ignored. It should be noted that MLP includes two intrinsic factors, namely, human face and body, so that it is not solely using extrinsic factors.

- *CEMNet w/t AMNet*: This is the proposed baseline model where both pipeline 1 (MLP) and pipeline 2 (AMNet) are used to jointly predict event memorability.

- *CEMNet w/t ICNet*: This is a variant of our baseline model where ICNet [45] is used in the place of AMNet in pipeline 2. This is to verify if our model is dependent on the specific CNN model used.

Our models are trained (with Adam optimizer) and evaluated on Pytorch platform using a machine with NVIDIA GeForce GTX 1080 GPU. For MLP, AMNet and ICNet, the batch size are set as 64, 16 and 32 respectively; learning rate are set as 0.001, 0.0001 and 0.0001; training epochs are set as 100.

### 5.1 Extrinsic features has higher predictive power

Table 2 shows the results of different configurations. The two CEMNet variants outperform the corresponding CNN-only benchmarks that use visual features only. For example, CEMNet w/t AMNet

| Method | Input Features | Precision↑ | Recall↑ | F1↑ | Mean error↓ |
|---|---|---|---|---|---|
| AMNet [15] | Image | 0.171 | 0.179 | 0.150 | 3.03 |
| ICNet [45] | Image | 0.153 | 0.155 | 0.140 | 3.11 |
| MLP | Extrinsic Features* | 0.389 | 0.385 | 0.333 | 0.91 |
| CEMNet w/t AMNet | Intrinsic + Extrinsic | **0.408** | **0.414** | **0.368** | **0.85** |
| CEMNet w/t ICNet | Intrinsic + Extrinsic | 0.369 | 0.340 | 0.340 | 0.97 |

Table 2: Comparing performance of models. *Intrinsic features, *i.e.*, human face & body, are included.

achieves an F1 score of $0.368$, which far exceeds the AMNet-only model with $F1 = 0.150$. Similar outcome is obtained from the ICNet variant with $F1 = 0.340$ on the full model and $F1 = 0.140$ on ICNet-only model. Such a pattern is also observed on the precision and recall metrics. Moreover, the CEMNet w/t AMNet gives slightly higher F1 score than the CEMNet w/t ICNet, possibly owing to more versatile visual features generated by its attention mechanisms. Meanwhile, MLP itself achieves an $F1$ score of $0.333$, which is more than double that of the CNN-based methods. This echos with results of linear regression analysis (Section 4.1), and clearly shows that extrinsic features are more important than intrinsic features in event memorability prediction. On the other hand, the intrinsic image features may still be useful in the sense that (1) both CNN-based models achieved above chance-level performance, and (2) they boost the performance of the two full models against MLP, leading to $10.5\%$ and $2.0\%$ increases in CEMNet w/t AMNet and CEMNet w/t ICNet, respectively.

Regarding the absolute mean error, our CEMNet w/t AMNet gives the best result of $0.85$. Note that a mean absolute error of $1.0$ indicates that on average the predicted memory score is one-step from the ground truth. Hence, with mean errors lower than $1.0$, the predictions made by the CEMNet variants, as well as the MLP, are reasonably good for decoding event memory. Also note that we have trained the respective benchmarking models (i.e., variants of CEMNet) using the current dataset, to make a fair comparison of their performance.

## 5.2 Ablation study

To further understand how different extrinsic/intrinsic factors contribute to event memorability, we conduct ablation study by excluding individual factors. The results are shown in Figure 3, where the $x$-axis shows the individual factors removed. A larger drop of F1 score compared to the respective full model (a horizontal dished line) indicates that a factor is important in the model. Consistent on three models, we observe a performance drop when one of the following factor are removed, namely, activity, place, face, boundary condition, test distinctiveness, and encode distinctiveness. Hence, these features are all conducive to predicting event memorability. The impact of three features are inconsistent on three models, including, human, treatment, and interval, where the F1 scores could be either higher or lower when these features are removed. It is therefore not conclusive regarding the effect of these factors. This could be caused by the fact that individual factors may co-vary with each other so that it is difficult to tease apart their contributions. This issue is further discussed in Section 6.

## 6 Discussions

The results of linear regression analysis provide evidence that event memorability is affected by intrinsic and extrinsic features. We did not make in-depth explanation on how individual factors affect event memorability as it is beyond the scope of the study. Some interesting insights are worth mentioning though. The results seem to support the event distinctiveness hypothesis but they do not support EST, since the correlation of the boundary condition is negative. There could be several reasons for this. First, the problem context of the EST is a bit different from our study. The events in EST are defined at more refined granularity, e.g., different stages of decorating a party room, whereas the everyday life events in our study have a coarser granularity. Second, the experimental setting is also different as EST experiments used videos to show the events, with a shorter encoding-testing interval compared to the current experiment.

We have hypothesized that it is possible to use multimodal input (intrinsic+extrinsic) features to predict item-wise event memorability. The CEMNet serves as a baseline model and gives evidence in support of this hypothesis. Meanwhile, there is a large space to improve the performance, which can be achieved in several ways: (1) using more complicated CNN structures in pipeline 1, (2) adopting novel image-based memorability prediction models in pipeline 2, and (3) designing more

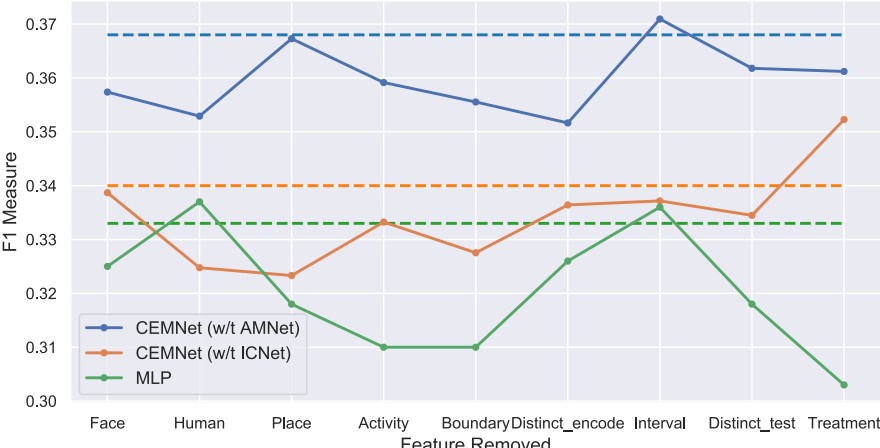

Figure 3: F1 score when a feature is removed. Dashed lines show the F1 scores of full models.

sophisticated mechanisms to fuse the output of two pipelines. This work focuses on the feasibility of the method using the novel R3 dataset. We leave the task of boosting the performance as future work. One may also try to extract additional features from our R3 dataset, or modify the experiment protocol to include other important intrinsic and extrinsic features. In any case, being able to reliably predict event memorability has important implications to computational cognitive studies.

We have also explored (through ablation study) the role of different visual semantics features in event memorability. We observed that five factors have consistent positive effects on the prediction of event memorability, although they have varying degrees of impact. However, this study does not provide information about the underlying causal relationship between a particular factor and event memory. Nor do we explore the combinations of visual semantics to achieve optimal prediction. This is related to one limitation of the current study, namely, there is possible co-variance of different intrinsic and extrinsic factors. Indeed, the multiple aspects of visual semantics may overlap with each other. For example, the presence of human face and body could be correlated; place may be indicative of the activity undertaken by a subject (and *vice versa*). Similarly, the encoding distinctiveness is contingent on the frequency of different activities. Hence, it may co-vary with the activity factor. Such co-variances not only jeopardize the stability of the models, but also make it difficult to precisely attribute event memorability to individual factors. This is shown by the lack of unanimous effect of individual factors in the ablation study. Future work may look into how to tease apart the effect of individual factors on event memory. Another limitation of the study is related to the profiles of human subjects. We only recruited elder adults, which was rooted in our goal of providing cognitive intervention to the at-risk population [52]. Hence, the findings may not generalize to alternative populations. Note also that there is possible gender bias due to the higher proportion of female subjects. In addition, the experiment used lifelog as a means of data collection, which may lead to privacy-related concerns as a potential negative social impact.

## 7   Conclusions

In this study, we propose to predict event memorability in the context of lifelogging and cued recall. We build the *first* large-scale dataset that consists of rich visual semantic features, event context related to memory encoding and test, and subjects' memory outcome. We design a simple yet effective event memorability prediction model to compute event memory scores from the visual and contextual features. This is the *first* work to perform item-wise event memorability prediction on contextual visual semantics (without using brain imaging data). The performance of the model at various configurations show the feasibility of event memorability prediction on visual semantics, and provides valuable insights into factors that contribute to event memorability. The proposed dataset and baseline model paves the way towards developing sophisticated methods to evaluate event memory, and further to design programs for cognitive intervention.

## Acknowledgments and Disclosure of Funding

This project is partially funded by the Singapore A*STAR JCO REVIVE Project (1335h0009).

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
