# A    Appendix

## A.1    R3 Experiment Protocol

### A.1.1    Population size

We conducted a pilot study with 8 young subjects and 5 older adults. Based on the power analysis, we set the target number of subjects as 40. Since this was a longitudinal study and there were expected data loss, we added a safety buffer and recruited 47 subjects in total.

### A.1.2    Sampling of daily events

For memory testing, we sampled about 10 events per day based on analysis of daily activity patterns of subjects. In essence, they were typically engaged in about 10 events that were clearly distinguishable and memorable. Notably, there are individual differences and variations across different days for an individual, depending on the subjects' life styles. So, 10 is an average number only. Nevertheless, an average of 10 events is widely adopted in the literature. For example, in similar settings of neuropsychology studies (memory research using lifelogging) [24, 25, 39], the sampling rate of events is set at 10 events per day.

Given that each testing session had a collection of data during 2 weeks, the total number of image cues is about 140. We set it at exactly 144 to accommodate the structure of testing protocol, which is elaborated next.

### A.1.3    Memory test and mnemonics

We test subject's event memory according to a graded 10-level mnemonics scheme. Following the "remember-know" paradigm [40], the memory test is conducted as follows.

A sequence of 144 image cues were shown one-by-one. These 144 images were equally split into 8 runs, each consisting of 18 trials (image cues) with one stimulus in each trial. There was a 2-minute break between two runs for the subject to rest. Among the 18 stimuli, 6 were drawn from the trained condition, 6 from the non-trained condition, and 6 were lure images. An image cue was displayed on the screen for 8 seconds. Next, a question prompt was shown with three options: Remember (R), Know (K) and New (N) (Figure 1(a)). The prompt remained on the screen for 3 seconds during which the subject pressed a button box to choose one option. Next, the second question prompt appeared. The content was contingent on the answer to the first question. For an 'R' answer in Q1, numbers buttons 1, 2, 3, 4 were shown, which corresponded to the level of remembered details, where 1 means few details and 4 means many details. For a 'K' answer, number buttons 1, 2, 3, 4 were shown, corresponding to the level of familiarity levels, 1 - low familiarity and 4 - high familiarity. For an 'N' answer, two buttons 'Perhaps' and 'Certain' appeared, which indicated how confident the subjects was regarding the 'New' decision. If the subject did not answer Q1, a blank (grey) screen was shown as place holder for Q2, and no action was required. Immediately after Q1, a new image cue appeared to start a new round of test. On average, each run lasted about 5 minutes, and the total testing session was about 50 minutes.

Subjects were given detailed instructions on the process before the testing and went through adequate trials. Corresponding to the subject's responds, the memory outcome was coded in a 10-level graded mnemonics for a image cue from one's own lifelog (Figure 1(b)). In the final dataset, the testing outcome of lure images were not included since they were not relevant to the subjects' memory. They were used for quality control purpose, *per se*.

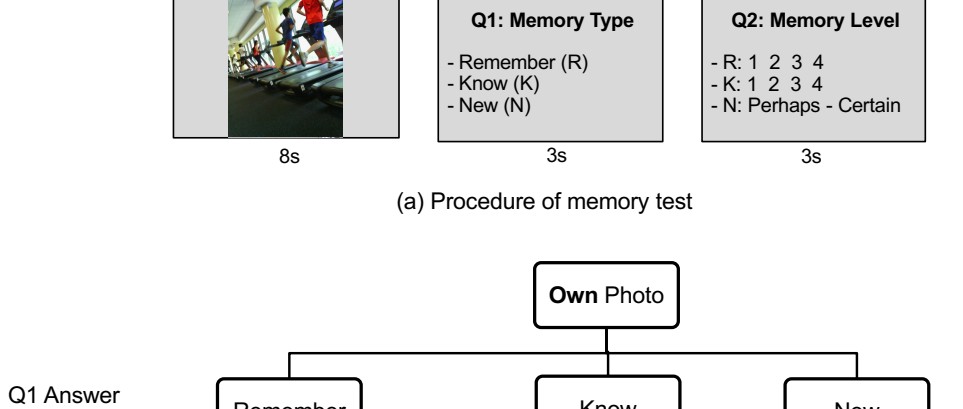

(a) Procedure of memory test

(b) Encode test outcome as 10-level graded memory types

Figure 1: Memory test protocol and memory mnemonics.

### A.1.4 Quality control and individual differences in memory tests

The subjects were in general quite accurate in identifying photos of their own life (accuracy > 80%) and rejecting lure images (accuracy > 90%). We do observe individual differences in the participants, e.g., some achieved higher hit rates than others; and the distribution of different memory categories were dissimilar. We have adopted a few exclusion criteria to ensure the data was of high quality.

- Absence of answers in 3 or more (out of 18) stimuli prompts in a run => exclude run.
- Hit rate (including remember and know responses)<50% and/or CR(correct rejection)<50% in a run => exclude run.
- Hit rate<60% and/or CR (correct rejection)<60% in a run => exclude run.
- Absence of 'Remember' or 'Know' answers in a run => exclude run.

### A.1.5 Metrics of event memory

In this work, we used ordinal numbers to represent event memorability score due to the following considerations. Previous works on image/video memorability use a different protocol to evaluate the memorability. In essence, the memorability score of an image/video is derived from the statistics of multiple responses from a number of subjects, making it naturally a real number between 0 and 1. In contrast, the episodic events memory studied in this research is highly personalized. Each individual image receives only one response (a discrete memory score) from a subject. So, the resolution of the memorability score is at best 0.1 (scaled to [0,1]). Moreover, there is neuro-cognitive basis for considering the memorability scores as ordinal numbers. In cognitive psychology, there are mixed evidences on the nature of event memory. On the one hand, researchers adopt graded strength of event memory as a continuous metrics [27]. On the other hand, many works [2, 23, 40, 47] show distinctive cognitive processes of "remember" and "know", which makes a continuous score of memory strength questionable. Therefore, we believe that there is no conclusive evidence to indicate if event memory is categorical or continuous. As a plausible strategy, we treat it as ordinal numbers; and accordingly,

| Activity Types | Place Categories |
|---|---|
| Child care, Cooking, Cycling, Dancing, Drawing, Calligraphy, Driving, Eating/drinking, Gardening, Gym exercise, Housekeeping, Making payment, Meeting, Pet care, Photo-taking, Playing badminton, Playing cards/mahjong, Playing golf, Presentation (Giving), Presentation (Listening), Reading, Running/jogging, Shopping, Swimming, Taking bus, Taking train, Talking, Using computer, Using phone, Walking, Watching movie, Watching TV/video, Yoga, Other | Airport terminal, Amusement park, Badminton court, Bakery, Bus, Bus stop, Car, Car Park, Casino, Church, Classroom/lecture theater, Clinic, Concert hall/stage, Conference room, Convenience store, Corridor, Dance studio, Department store, Electronic store, Exhibition hall, Fitness corner, Food court, Food stall, Functional room, Gym, Hair salon, Home, Hospital, Library, Mall, Natural scene, Office, Pharmacy, Plane, Playground, Restaurant/Cafe, Shop, Street/Pavement, Supermarket, Train, Train station, Void deck, Wet market, Yard/Garden, Other |

Table 1: List of activities and places as extrinsic context.

we have carefully chosen "ordinal categorical cross-entropy" as the loss function when training the models.

For performance evaluation, our model generates ordinal numbers as the memory scores, whereas we use MAE as the aggregated score of multiple events and multiple subjects to show the overall performance. When we trained the CEMNet model, we also tried to use mean squared error (MSE) as the loss function, which gave slightly worse results. In reporting the performance of different models, we need to aggregate results of different memory categories and across multiple splits. Therefore, mean average error (MAE) is reported as a preferred performance metric. The MAE value, while related to the level of the ordinal category, should not be interpreted as the ordinal category, *per se*.

## A.2 Ethical issues

Subjects were compensated for their participation in the data collection experiment. They were paid $40 for each experiment session (including 2-3 days of lifelog device usage and a visit to the experiment site). They typically participated in a total of 10 sessions, so that that total payment was about $400. Informed written consent was obtained from all subjects before the start of the experiment. The lifelog data may contain personal identifiable information, namely, human faces. Based on the consent agreement, such information are removed (e.g., by pixelating the facial regions). The study was approved by the Institutional Review Board (IRB) of the National University of Singapore. All methods were performed in accordance with the relevant guidelines and regulations.

## A.3 List of activities and places

Two aspects of extrinsic features are activities of the subject and places when an event happens (captured by lifelog photo). Table 1 shows the full list of activity types and place categories. The activities are specific to the cohort of elder people and hence is more semantically relevant to their life styles. Note that we do not intend to provide an exhaustive list of human activities. For a set of non-frequent activities not in our list, we annotate them as "other". If a photo cue show ambiguous activity, the annotator backtracked to the original collection of lifelog photos to get precise annotation. Similar comments can be made on the list of places.