# OpenReview forum: "Predicting Event Memorability from Contextual Visual Semantics"
_NeurIPS.cc/2021/Conference — NeurIPS 2021 Poster_

### Official Review · Reviewer_ZmrY · 2021-07-12

**Rating:** 6
**Confidence:** 3

**Summary:**

In service of studying event memorability, the authors construct a
corpus of 10.6K images collected by 47 elderly study participants. The
collection process involves the participants wearing cameras that take
photographs at the rate of 2 frames per minute, and then regularly
checking-in with researchers who gather memorability ratings from the
participants for each image. The authors explore a variety of
intrinsic (i.e., having to do with the image itself) and extrinsic
(i.e., having to do with how the image relates to other images in the
participant's log) feature sets for predicting the ratings. They find
that extrinsic factors are the best predictors, and that adding
additional CNN image features adds only a bit of extra prediction
capacity.

**Limitations And Societal Impact:**

See above for some discussion --- I thought the authors did a good job, overall. All my checkboxes are checked: IRB approval for human subjects research, qualification of model limitations, and discussion of potential harms in applying models trained on the dataset.

**Main Review:**


The main contribution of this work, in my view, is the corpus itself:
the authors do an excellent job of making the case for the collection
of this type of data, and clearly are involved with/invested in the
outcome of memorability research for the benefit of elderly
folks. This dataset could become an excellent resource for the ML
community, both for use-cases having to do with memorability
prediction and beyond.

Technically, the authors do a good job of of exploring the corpus, and
undertake an admirable feature engineering effort to tease out what
makes a particular frame memorable versus not: extrinsic features
explored include auto-segmentation timing, relation of a frame to all
other frames, etc. Intrinsic features explored include: previous
memorability models trained on different corpora, hand annotations for
presence of faces, etc. Overall, I have no major technical concerns:
while this paper is less focused on modeling contributions (I suspect
more modern neural architectures could push performance at least a
bit), the dataset/data exploration contributions are the centerpoint
(along with reporting reasonable baselines).

The authors are straightforward in their discussion of shortcomings;
the ablation study in Figure 3 does indeed suggest that the patterns
different models use to achieve accurate predictions are not
straightforward. While this does cast some doubt on whether or not the
correlations discussed are meaningful in a more causal sense, the
authors mention this directly. The authors also correctly point out
that the gender and age distribution of their corpus may not be
representitive for many use cases.

My concerns include:

- While the authors do a per-feature analysis to demonstrate which are
  most predictive of memorability, I would have appreciated a more
  direct connection with the data. For example, they hand-coded for
  various activity types in each of the 10K frames and found a
  significant correlation, but which events types were memorable, and
  which were not? Perhaps table 1 could be augmented with this type of
  direct connection to the dataset.

- It should perhaps be noted more explicitly/earlier that the authors
  purposefully make no attempt at personalization: their
  cross-validation splits hold out entire participants, rather than
  individual images. While this is a fine decision, I would have liked
  a note earlier on the rationale for measuring globally memorable
  events (i.e., events that are likely to be memorable to people, on
  average) vs. personally memorable events. Potentially promising
  avenues of future work would be: 1) exploring deviations in the
  predictions of an "impersonal" model vs. a personal one; and 2) how
  to possibly train personalized models.

- Given the strong performance of an MLP on extrinsic features
  reported in Table 2, I would have been curious to see the
  performance of a linear model, whose coefficients may provide more
  direct hints at the correlations between input features and the
  label. Another more interpretable alternative might be the
  Generalized Additive Models; a nice library for them is:
  https://github.com/interpretml/interpret

  @article{nori2019interpretml,
    title={InterpretML: A Unified Framework for Machine Learning Interpretability},
    author={Nori, Harsha and Jenkins, Samuel and Koch, Paul and Caruana, Rich},
    journal={arXiv preprint arXiv:1909.09223},
    year={2019}
  }

**Time Spent Reviewing:**

2

---

> ### Author Response · Authors · 2021-08-08
> **Answers to Reviewer ZmrY**
>
> === Detailed analysis of the relationship between individual factors and memory outcome ===
> This is an excellent suggestion (also mentioned by reviewer 9z5U. Please refer to my previous answers). In essence, we choose not to report this due to page limits, as well as to stay focused on the current scope of work. We will add it as an item for future work in the discussion section.
>
> === Early mentioning of “no attempt at personalization”===
> Yes. We will add a note in the introduction
>
> === Suggested future work ===
> We will add these in the revision.
>
> === More in-depth discussion of linear regression model and adopting generalized additive models===
> Thanks for the excellent suggestion! We will conduct more experiments using a linear model/generalised additive models in future and report the result, along with other interesting findings stated earlier (e.g., interpretation of the effect of individual factors on memory).

---

### Official Review · Reviewer_NWjs · 2021-07-15

**Rating:** 8
**Confidence:** 3

**Summary:**

EDIT AFTER AUTHOR RESPONSE: I read the other reviews, and I did not see the need to make any changes to my review.

This paper develops a novel experimental paradigm and introduces a dataset obtained from a lifelogging experiment where subjects rate how well they remember events corresponding to photos taken by first-person view photos.  It answers the question of whether intrinsic image features or extrinsic contextual factors are more predictive of memorability, and it further explores which factors are the most important for predicting memorability.

**Limitations And Societal Impact:**

Yes.

**Main Review:**

Originality:  High.  It appears that the experimental paradigm is novel.  The features extracted for the analyses seem to have been developed in prior work, and the network used is a minor variation of existing architectures.  But the analysis as a whole is fairly sophisticated and constitutes a non-trivial application of existing approaches.

Quality:  High.  The experiment appears to have conducted in a careful manner, and the analyses appear to be correct.

Clarity:  The paper is clearly written.

Significance:  Moderate.  Predicting event memorability may be relevant to many disciplines interested in user experiences, and this paper contributes very interesting findings which help us gain a better understanding of what factors make an event memorable.  Furthermore, the dataset could be useful in other studies.  The finding that photo reviews helps with memory retention is potentially clinically significant, but this particular finding is not the focus of this work.


**Time Spent Reviewing:**

2

---

> ### Author Response · Authors · 2021-08-08
> **Answers to Reviewer NWjs**
>
> Thanks for the positive feedback!

---

### Official Review · Reviewer_9z5U · 2021-07-18

**Rating:** 7
**Confidence:** 4

**Summary:**

This paper makes 3 contributions. First, the paper proposes and builds an event memorability dataset, wherein life-logging and subsequent event recollection data are collected from 47 elder subjects over a period of 3 months (Section 3.1). Next, the authors investigate several potential predictors of event memorability, including intrinsic image cues and extrinsic factors (Section 3.2-4.1). Lastly, the authors consider baseline models, in the form of a contextual event memory network to predict event memorability (Section 4.2 and 5).

**Limitations And Societal Impact:**

Yes, the authors have adequately addressed the limitations and potential negative societal impact of their work.

**Main Review:**

EDIT: I am happy with the authors' response and I am raising my score.

The presented paper is very promising -- the proposed idea of understanding and predicting event memorability is a novel topic and will be a welcome addition to the memory literature (for both computational and psychology researchers). The authors have also put a remarkable amount of effort into the paper, including collecting a large-scale longitudinal dataset, and to me, the entire effort is very commendable. Thus, my impressions of the paper are largely positive and I think this paper can be a nice addition to Neurips (or a similar high-impact venue).

That being said, I have a number of reservations about the paper in its current form. To me, the paper needs significant polish and improvement in the structure of the paper in order for this work to meet the promise it has. Thus, my current rating is borderline (I would be open to improving my score based on the rebuttal). Below are my suggestions and concerns which I hope the authors will find helpful.

(a) First, I would like to see more clarity and details about the procedure used by the authors to collect the dataset (Section 3.1). Some questions I would like to see addressed are below.

(i) What was the rationale for collecting 47 subjects? Did the authors do any power analysis or any pilot experiments to determine how many subjects they are going to collect?

(ii) In the memory test, subjects were presented with 144 images one after another. What was the rationale for deciding this number i.e. presenting 144 images? Did you take any steps to ensure subjects will not ``bored" during the task? To me, 144 seems like a large number of trials presented in one session; could you also provide information on how long the memory test takes for a subject on average? I think an alternative way to collect data would have been to recruit a larger subject pool (but having fewer images per subject) -- did the author consider this possibility? Basically, my questions here seek to understand the rationale behind the data collection procedure..

(iii) Btw, were the subjects informed that some images during the memory test will be "lure" images? I am asking this because not informing subjects about the experimental manipulation could have significant consequences on the data collection.

(b) After reading Section 3.1, I was really craving to know a bit more about the dataset that the authors collected and I was rather surprised that the paper didn't include details about the same. For instance, how accurate were the subjects in remembering the events? Were there any interesting individual differences in the participants? Were there any participants that did particularly well or particularly poorly? (I know that the authors removed the data where subjects performed poorly but I think more analysis about subjects' performance on the dataset would provide valuable insights).

(c) Section 3.3 -- The authors here have done a commendable job in coming up with extrinsic features based on extant cognitive theories and then using these features (along with intrinsic cues) to better understand event memorability (Section 4.1). However, I think that this section can be significantly strengthened and its structure can be substantially improved as suggested below.

(i) In Section 3.3.1, the authors considered two extant cognitive theories, event distinctiveness, and event segmentation theory but it was unclear to me why the authors considered these two theories and not any other theories -- are these the two most dominant theories in cognitive science literature? Are there any other possible theories that the authors could have considered (but perhaps didn't because of computational constraints)? Perhaps, this section could be improved if the authors provide a brief overview of existing theories of event memory and highlight why they decide to consider event distinctiveness and event segmentation.

(ii) In Section 4.1, from the results presented in Table 1, the authors conclude that the most important features include image memorability, presence of faces, event distinctiveness at encoding, activity, treatment, and encode-test interval. I think here readers familiar with cognitive research will really appreciate a more detailed discussion of the implications of these findings -- do these findings provide insights to the event memory literature? From my understanding (please correct me if I am wrong), these results provide evidence to the event distinctiveness hypothesis but they do not support the event segmentation theory (since the correlation of the boundary condition is negative). Do the authors have any hypothesis/insights on the same or why the boundary condition did not perform as expected? My point here is that Section 3.3 and 4.1 can go beyond showing event memorability can be predicted by intrinsic and extrinsic cues. Rather, I think this section provides a great opportunity to provide valuable insights to cognitive research as the large-scale dataset and the computational methods developed here can be used to evaluate existing theories of event memory.

Minor points

(a) The caption of Table 1 says that t-value in bold font means the factor significantly correlated with memory (p > 0.05). Shouldn't it be p < 0.05?

(b) Line 48: I would refrain from using strong words such as "convincing" evidence. I think simply stating that you provide evidence should suffice as the results speak for themselves.

**Time Spent Reviewing:**

5

---

> ### Author Response · Authors · 2021-08-08
> **Answers to Reviewer 9z5U**
>
> === Sample size and pilot study ===
> Yes, we have conducted a pilot study with 8 young subjects and 5 older adults. Based on the power analysis, the target number was 40 subjects. Since this was a longitudinal study and there were expected data loss, we added a safety buffer and recruited 47 subjects.
>
> === Details on experiment protocol.===
> First, the testing images are typically extracted from lifelog data collected in about 2 weeks (Refer to the data collection protocol L98-101). With an average of 10 events per day, the total number of image cues is about 140. Please refer to line 188 and feedback to reviewer wyti for the rationale of adopting 10 events per day. Also refer to [22,23,33] for similar settings. We set the total number of stimuli as 144 to accommodate the requirements of the testing protocol. Specifically, these 144 images are equally split into 8 runs, each consisting of 18 trials (image cues) with one stimulus in each trial. There is a 2-minute break between two runs for the subject to rest. Among the 18 stimuli, 6 were drawn from the trained condition, 6 from the non-trained condition, and 6 were lure images. On average, each run lasts about 5 minutes, and the total testing session is about 50 minutes. We will add these details in the Appendix.
> Recruiting more subjects while reducing the number of trials per subject is a possible alternative. However, the cost of recruiting more subjects is high and we chose to make the best use of individual subjects without being too taxing on them.
>
> === Subjects are aware of lure images ===
> We explained this clearly to the subjects before the testing and conducted warm-up trials to get them familiar with the protocol.
>
> === Quality control and individual differences in memory tests ===
> We tried to keep the description of data collection succinct due to page limits.
> The subjects were in general quite accurate in identifying photos of their own life (accuracy > 80%) and rejecting lure images (accuracy > 90%).
> Yes, there are individual differences in the participants, e.g., some achieved higher hit rates than others; and the distribution of different memory categories were dissimilar. We have adopted a few exclusion criteria to ensure the data was of high quality (L130-132).
> 1.	Absence of answers in 3 or more (out of 18) stimuli prompts in a run => exclude run.
> 2.	Hit rate (including remember and know responses) < 50% and/or CR (correct rejection) <50% in a run => exclude run.
> 3.	Hit rate <60% and/or CR(correct rejection) <60% in a run => exclude run.
> 4.	Absence of 'Remember' or 'Know' answers in a run => exclude run.
>
> We would like to add these in the appendix.
>
> === Alternative cognitive theories regarding extrinsic features. ===
> There are alternative theories based on which extrinsic factors can be extracted, such as, memory re-instatement [23, 26], passage of time [McGaugh, 2000], emotion [Daselarr et al, 2008, Piolino et al, 2008], attention [Cowan, 1998]. Among them, memory re-instatement and passage of time have been considered in the current work using “treatment” and “interval” factors, respectively. Emotion and attention are not included because of the difficulties in evaluating them computationally under the current experiment protocol. To the best of our knowledge, the two theories adopted (“event distinctiveness”, “event boundary condition”) are most relevant to the study and they are prevalent in the literature. Therefore, we have adopted them.
> 1.	J. L. McGaugh. Memory–a century of consolidation. Science, 287(5451):248–251, 2000. 2
> 2.	Daselaar, S. M. et al. The spatiotemporal dynamics of autobiographical memory: neural correlates of recall, emotional intensity, and reliving. Cereb. cortex 18, 217–229 (2008).
> 3.	Piolino, P. et al. Reliving lifelong episodic autobiographical memories via the hippocampus: A correlative resting pet study in healthy middle-aged subjects. Hippocampus 18, 445–59 (2008).
> 4.	N. Cowan. Attention and memory: An integrated framework. Oxford University Press, 1998. 2
>
> === Detailed analysis of the relationship between individual factors and memory outcome ===
> Thanks for the good suggestion! Indeed, this is an issue we wish to address but skipped in the current paper due to (1) page limits, (2) scope of work. In fact, we have discovered some preliminary, interesting results on how individual intrinsic and extrinsic factors contribute to the event memory. However, to keep the current work focused (on a novel dataset and item-wise event memorability prediction), we decided to defer this part to another potential publication.
>
> === Possible reasons that the current results did not support EST ===
> There could be several reasons why the current results do not support event segmentation theory (EST). First, the problem context of the EST is a bit different from our study. The events in EST are defined at more refined granularity, e.g., different stages of decorating a party room, whereas the everyday life events in our study have a coarser granularity. Second, the experimental setting is also different as EST experiments used videos to show the events and there is a short span between encoding and testing. In contrast, our study featured a longer encode-test interval and involved more complicated life events.
> Anyway, we appreciate the constructive feedback! We believe it is worthwhile to conduct further analysis with detailed discussions to provide more insights on this issue (among others). However, due to page limits, we may only be able to do so in a separate paper.
>
> === Typo in table 1.===
> Yes. It should be p<0.05.
>
> === L48 remove strong words (“convincing” evidence) ===
> Thanks! We will tone down the statement.

---

> > ### Comment · Reviewer_9z5U · 2021-08-10
> > **Response to rebuttal**
> >
> > Dear Authors,
> >
> > Thank you for your detailed response, I have raised my score based on your response. Below are some comments:
> >
> > ==Regarding Sample size and pilot study, and Subjects being aware of lure images==
> > Thank you for clarifying these. I would recommend mentioning this in the main paper or the Appendix -- it will further strengthen your paper as they show that the behavior experiments are rigorous.
> >
> > === Alternative cognitive theories regarding extrinsic features. ===
> > Thanks for clarifying this. If space allows, I would recommend adding these as some readers might appreciate this background and your rationale behind the theories you evaluate.
> >
> > === Possible reasons that the current results did not support EST ===
> > Thanks for clarifying this. Space permitting, could you add this in the paper perhaps in the Discussion section? Readers will appreciate your insights. For future work, I believe that the authors could use this dataset and the computational framework to evaluate extant theories of memory and perhaps submit that to a psych venue as this kind of work will also benefit the cognitive science community at large.

---

### Official Review · Reviewer_wyti · 2021-08-03

**Rating:** 7
**Confidence:** 4

**Summary:**

The paper explores episodic memory and answers questions related to the impact of context and the influence of intrinsic image properties. They gathered data through an experiment with 47 elder subjects over 3 months, which led them to build a a novel dataset for event memorability. This allows them to build CEMNet, a network that predicts event memorability at the item level.



**Ethical Concerns:**

No ethical concerns.

**Limitations And Societal Impact:**

The limitations are correctly outlined in Section 6. One main limitation is related to the choice of population for the study (elder subjects), which makes the results less generalizable to broader populations.

**Main Review:**

### UPDATED SCORE FROM 6 TO 7 BASED ON AUTHOR REBUTTAL.

## General thoughts
The paper sheds new light on a field that is seldom explored, episodic memory and the possibility of predicting it. The data collected is valuable. In general, the experimental approach to collect the data and evaluate the results is solid. I have a few concerns related to the predictive approach and models: the losses used appear to be a bit inconsistent, the model is somewhat simplistic, and there is little comparison to other models. In general, the paper is interesting but could be improved on the modeling side.

## Pros
Solid paper tackling a problem that hasn't been thoroughly addressed before.
In-depth experiment with live participants. The data collected has significant value.
Solid description of experimental protocols and methods. There is care in the development of the experimental procedure.
Interesting predictive approach.
First paper that studies episodic memory in this way.

## Cons
l. 27: there is a mention to understanding event memory beyond simple image recognition where [20] and [8] are cited. These works focus on visual working memory and not episodic memory - I'm not sure if they are the right citations there.

Missing citations on video memorability, arguably more connected to episodic memory than image memorability:

- Newman, A., Fosco, C., Casser, V., Lee, A., McNamara, B. and Oliva, A., 2020. Multimodal memorability: Modeling effects of semantics and decay on video memorability. ECCV 2020
- Cohendet, R., Demarty, C.H., Duong, N.Q. and Engilberge, M., 2019. VideoMem: constructing, analyzing, predicting short-term and long-term video memorability. ICCV 2019

There is one main concern: the dataset is built with 10k images from 40 participants, with the objective of creating a large database that can be used to train models. Episodic memory is, however, extremely user-dependent; this makes training a model with images coming from different users an ill-posed problem. How was this addressed?

l. 283: The ordinal nature of the memorability score is not well justified: why can't it be treated as a real number between 0-1 like in previous image/video memorability works? Why was "ordinal categorical cross-entropy" used instead of MSE? If an order-aware loss was the idea here, were there attempts to use a differentiable ranking loss like [1] or [2]?

l. 296: here, you compute performance with MAE, which speaks of a real-valued memory score: why is it referred to as ordinal above? There might be an inconsistency between training and evaluation if the order is used as supervision during training but MAE is used during evaluation.

The ablation studies are not super consistent: why are you seeing such differences between each model?

There are few comparisons to other works or models in the modeling sections. What happens if you train a MemNet or a SemanticNet on this data (specifically for the visual pipeline)?

## References:
[1] Blondel, M., Teboul, O., Berthet, Q. and Djolonga, J., 2020, November. Fast differentiable sorting and ranking. In International Conference on Machine Learning (pp. 950-959). PMLR.

[2]  Engilberge, M., Chevallier, L., Pérez, P. and Cord, M., 2019. Sodeep: a sorting deep net to learn ranking loss surrogates. In Proceedings of the IEEE/CVF Conference on Computer Vision and Pattern Recognition (pp. 10792-10801).

## Questions

- l.158: There are multiple other ways to measure image distinctiveness (e.g. KDE-based techniques). Were other methods attempted?

- l. 246 comparison with Image Memorability: were those models pretrained on image memorability datasets first?

- l. 248 why is pearson correlation used and not spearman?

- l. 313: what is your learning rate schedule?

- Calculation of event boundary condition: can you give more context about why use 10 events per day? Why is it considered normal? how does that impact your results?

**Time Spent Reviewing:**

10

---

> ### Author Response · Authors · 2021-08-07
> **Answers to Reviewer wyti**
>
> ===Adding references on video memorability===
> These are interesting citations. Based on our investigation, although videos (compared to still images) may have closer connection to episodic memory, there are fundamental differences between third-person-view video (and image) memory and the first-person-view real-life event memory as studied in the current work. Importantly, real-life events are of personal relevance, strongly contextualized in a myriad of other life events. As such, video memorability is more closely related to image memorability than to event memorability, adopting similar experiment protocol (Newman et al, 2020; Cohendet et al. 2019) as image memorability, except that they use short videos instead of images. In comparison, the protocol of our study is notably different from both streams of the above works. Nevertheless, these works are valuable references and we will include them in the revision.
>
> ===How do we handle individual differences?===
> Indeed the episodic memory is user-dependent, which is one of the main challenges to build a predictive model of event memory. This is also the main motivation of this study to use a suite of intrinsic and extrinsic factors to predict the memory. Importantly, the extrinsic factors aim to partially capture user-specific information, such as, activities carried out by the user and places visited, etc. We also included user demographic information (e.g., age, gender, education level, etc.) in the feature set. Combining all these factors, we expect that our model can account for individual differences, which is validated by the experimental results.
>
> ===Adopting ordinal number as metrics of event memory===
> Previous works on image/video memorability use a different protocol to evaluate the memorability. In essence, the memorability score of an image/video is derived from the statistics of multiple responses from a number of subjects, making it naturally a real number between 0 and 1. In contrast, the episodic events memory studied in this research is highly personalized. Each individual image receives only one response (a discrete memory score) from a subject. So, the resolution of the memorability score is at best 0.1 (scaled to [0,1]). Moreover, there is neuro-cognitive basis for considering the memorability scores as ordinal numbers. In cognitive psychology, there are mixed evidences on the nature of event memory. On the one hand, researchers adopts graded strength of event memory as a continuous metrics [27, Johnson et al. 2015]. On the other hand, many works [2,21,34,41] show distinctive cognitive processes of “remember” and “know”, which makes a continuous score of memory strength questionable. Therefore, we believe that there is no conclusive evidence to indicate if event memory is categorical or continuous. As a plausible strategy, we treat it as ordinal numbers; and accordingly, we have carefully chosen “ordinal categorical cross-entropy” as the loss function when training the models. For performance evaluation, our model generates ordinal numbers as the memory scores, whereas we use MAE as the aggregated score of multiple events and multiple subjects to show the overall performance.
> Johnson, M. K., Kuhl, B. A., Mitchell, K. J., Ankudowich, E. & Durbin, K. A. Age-related differences in the neural basis of the subjective vividness of memories: Evidence from multivoxel pattern classification. Cogn. affective & behavioral neuroscience 15, 644–661 (2015).
>
> When we trained the CEMNet model, we also tried to use MSE as the loss function, which gave similar, but slightly worse results. In reporting the performance of different models, we need to aggregate results of different memory categories and across multiple splits. Therefore, MAE is reported as a preferred performance metric. The MAE value, while related to the level of the ordinal category, should not be interpreted as the ordinal category, per se.
>
> Thanks for the citations (Blondel et al., 2020; Engilberge et al. 2019) on how to deal with ordinal numbers in model training. As mentioned, we adopt “ordinal categorical cross-entropy” (from Keras library) mainly due to the cognitive plausibility. We will consider these citations for further analysis in future.
>
> === Inconsistent results in ablation study ===
> First, for the challenging problem of predicting event memorability, there are considerable uncertainties in the variables and their relation to the memory outcome. Therefore, the result is expected to be unstable to a certain extent. Thus, random errors exist, which caused the inconsistencies in the ablation study. Second, as explained in the discussion section (Lines 360-369), there could be co-variance of different intrinsic and extrinsic factors, making it difficult to tease apart their effects. This is especially true when we use DCNN models, which are largely black-boxes whose behaviors are difficult to explain. In the discussion, we acknowledge this as a limitation and leave it for future work.
>
> === Re-training on the current dataset for benchmarking methods.===
> Actually, we trained the respective benchmarking models (i.e., variants of CEMNet, section 4.2) using the current dataset. The outcome is reported in section 5. We did mention in Section 4.1 (lines 258-260), that the data used in linear regression model was not re-trained on our dataset. We will state explicitly in Section 5 on the training strategy in the revision.
>
> In the linear regression analysis, the models are pre-trained on their original dataset without re-training on our dataset. Here, the analysis was intended to get a rough estimation of the model capability. In the subsequent item-wise memorability prediction (section 4.2, and section 5), we re-trained the model on our dataset.
>
> === Alternative ways to compute image distinctiveness.===
> This is an excellent suggestion! For now we computed the event distinctiveness based on cosine similarity of images; and have not attempted alternative methods. We will investigate if other methods (e.g., KDE-based) give different outcomes in future.
>
> === Using Pearson correlation analysis ===
> The predicted memory scores produced by the legacy models are real numbers whereas the ground truth memory scores from our dataset are ordinal number. Spearman correlation analysis reduces the power of the prediction. Specifically, due to the low resolution of ground truth memory scores (ordinal numbers), there are many ties when doing the ranking, which makes the computation of p-value inaccurate. Nevertheless, we have also computed Spearman’s correlation, and observed similar results as Pearson.
>
> === Learning rate schedule ===
> We did not use any learning rate schedule during training, i.e., the learning rates are fixed: AMNet lr=0.001, MLP and IC_NET: 0.0001.
>
> === Why do we consider 10 events per day as normal?===
> We decided to use 10 events per day based on analysis of daily activity patterns of subjects. The rationale being that subjects typically engage in about 10 events per day that are clearly distinguishable and memorable. Notably, there are individual differences and variations across different days for an individual, depending on the subjects’ life styles. So, 10 is an average number only. Nevertheless, an average of 10 events is widely adopted in the literature. For example, in similar settings of neuropsychology studies (memory research using lifelogging) [22,23,33], the sampling rate of events is also about 10 events per day.

---

> > ### Comment · Reviewer_wyti · 2021-09-03
> > **Response**
> >
> > Thanks for your response. This clarifies most of my concerns. The ordinal approach for scores makes sense, and I am more convinced about the workaround regarding individual differences. I would love to see how this model performs with increased hyperparameter search in a follow-up. Adding the missing references and clarifying the discussed points in the final version should make the paper pass the NeurIPS bar. I advocate for an acceptance here. I am changing my score from 6 to 7.

---

### Decision · Program_Chairs · 2021-09-27

**Decision:**

Accept (Poster)

**Comment:**

The paper studies episodic memory in humans and presents a novel event memorability dataset, along with a memory network (CEMNet) to enable computational studies. All the reviewers are positive about the paper and the authors’ responses addressed most of their major concerns. I strongly encourage the authors to make the suggested changes  by reviewers — better description and analysis of the data collected and the process used (ZmrY, 9z5U), writing clarifications to sections 3 and 4 (9z5U, ZmrY), adding ablation studies (wyti, ZmrY) and updating the references (wyti, ZmrY).